

# Global sensitivity analysis of the GEOS-Chem chemical transport model: Ozone and hydrogen oxides during ARCTAS (2008)

Kenneth E. Christian[1], William H. Brune[1], and Jingqiu Mao[2,3]

[1]Department of Meteorology and Atmospheric Science, Pennsylvania State University, University Park, PA (USA)
[2]Program in Atmospheric and Oceanic Sciences, Princeton University, Princeton, NJ (USA)
[3]Geophysical Fluid Dynamics Laboratory/National Oceanic and Atmospheric Administration, Princeton, NJ (USA)

*Correspondence to:* Kenneth Christian (kec5366@psu.edu)

**Abstract.** Developing predictive capability for future atmospheric oxidation capability requires a detailed analysis of model uncertainties and sensitivity of the modeled oxidation capacity to model input variables. Using oxidant mixing ratios modeled by the GEOS-Chem chemical transport model and measured on the NASA DC8 aircraft, uncertainty and global sensitivity analyses were performed on the GEOS-Chem chemical transport model for the modeled oxidants hydroxyl (OH), hydroperoxyl (HO$_2$), and ozone (O$_3$). The sensitivity of modeled OH, HO$_2$, and ozone to modeled inputs perturbed simultaneously within their respective uncertainties were found for the period of NASA's Arctic Research of the Composition of the Troposphere from Aircraft and Satellites (ARCTAS) A & B campaigns (2008) in the North American Arctic. For the spring deployment (ARCTAS-A), ozone is most sensitive to the photolysis rate of NO$_2$, the NO$_2$ + OH reaction rate, and various emissions, including methyl bromoform (CHBr$_3$). OH and HO$_2$ were overwhelmingly sensitive to aerosol particle uptake of HO$_2$ with this one factor contributing upwards of 75 % of the uncertainty in HO$_2$. For the summer deployment (ARCTAS-B), ozone was most sensitive to emissions factors, such as soil NOx and isoprene. OH and HO$_2$ were most sensitive to biomass emissions and aerosol particle uptake of HO$_2$. With modeled HO$_2$ showing a factor of 2 underestimation compared to measurements in the lowest 2 kilometers of the troposphere, lower uptake rates ($\gamma_{HO_2} < 0.04$), regardless of whether or not the product of the uptake is H$_2$O or H$_2$O$_2$, produced better agreement between modeled and measured HO$_2$.

# 1 Introduction

With rising temperatures, shrinking sea ice, and expanding emissions into the atmosphere from increased human development and biomass burning, the Arctic is experiencing rapid changes felt nowhere else on the globe. While the region is largely undeveloped, anthropogenic air pollution from Northern Hemisphere population centers in East Asia, Europe, and North America is regularly advected into the Arctic atmosphere, contributing to the "Arctic haze", (e.g., Barrie et al., 1981). Increasing oil and gas exploration and extraction, coupled with summertime shipping lanes through the region will make air pollution worse. A better understanding of atmospheric oxidation chemistry is needed in order to provide a scientific basis for a sound mitigation strategy to combat this likely deteriorating air quality.

Atmospheric oxidants are at the forefront of any air chemistry study because the lifetimes of most gaseous and particulate species are determined through oxidant reactions. The primary atmospheric oxidizers of interest are the hydroxyl radical (OH),



the hydroperoxyl radical (HO$_2$), collectively referred to as HO$_x$ (HO$_x$ ≡ OH + HO$_2$), and ozone (O$_3$) (Levy, 1971) . Ozone, OH, and HO$_2$ are coupled in a cycle in which ozone photolysis leads to the creation of OH, which then cycles with volatile organic compounds to create HO$_2$, which then can react with nitric oxide (NO) to ultimately produce ozone and recycle OH. While this cycle appears to be well known and documented, models still fail in describing atmospheric composition (e.g., Wu et al., 2007). These model shortcomings are usually attributed to errors in the chemical reaction rates, emissions, or meteorology (e.g., Wild and Prather, 2006).

A useful tool for examining and attributing sources to these model shortcomings is sensitivity and uncertainty analyses. In performing sensitivity analyses, there are two basic approaches: local and global. Local sensitivity analysis involves varying model inputs one at a time around a given point in input space while holding all other model inputs constant. This method assumes at least locally linear input–output relationships. Global sensitivity analyses; on the other hand, involve the simultaneous perturbation of all the model inputs allowing for the interactions between inputs to be analyzed as well (Rabitz and Aliş, 1999). Global sensitivity analysis does not assume that the input and output have a linear local relationship and in fact can test the sensitivity of the output factors to the co-variation of two more input factors. Global sensitivity analysis is preferred over local sensitivity analysis for complex models (Saltelli et al., 2008) and applies well to global chemical transport models (CTMs), such as the GEOS-Chem (Goddard Earth Observing System-Chemistry) model used in this study, that can have non-linear interactions in the chemical kinetics, emissions, and meteorology.

Previous sensitivity studies using GEOS-Chem tended to use local sensitivity methods despite the known non-linearity of the underlying chemical processes and subsequent interactions with meteorological and emissions factors. To combat these non-linearities, a common strategy in sensitivity studies involves the perturbation of model factors across a smaller sample of the input space (e.g., Fiore et al., 2009; Wu et al., 2009). While useful in ascertaining sensitivities for individual factors, this method cannot provide a complete picture of the modeled uncertainty as the entire input space is not sampled. In other sensitivity studies GEOS-Chem has been analyzed for its sensitivity to meteorological models and factors (e.g., Wu et al., 2007; Heald et al., 2010) and both biogenic (Fiore et al., 2005; Mao et al., 2013b) and anthropogenic emissions (e.g., Fiore et al., 2002; Martin et al., 2003; Auvray and Bey, 2005; Jaeglé et al., 2005; Guerova et al., 2006). While helpful, these local sensitivity studies were limited to perturbing a small set of similar input factors so it is possible that some important input factors or interactions may have been missed.

This study covers National Aeronautics and Space Administration's (USA) (NASA's) Arctic Research of the Composition of the Troposphere from Aircraft and Satellites (ARCTAS) campaign (2008) (Jacob et al., 2010). The impetus of the campaign was to better understand the complex interactions between atmospheric composition, the environment, and climate in the North American Arctic and was split into three sub-campaigns, ARCTAS-A (spring), ARCTAS-CARB (California—not included in this study), and ARCTAS-B (summer). ARCTAS-A sought to better understand the chemical processes during the polar sunrise when anthropogenic pollution is at its annual maximum and halogen chemistry is active and was based in Fairbanks, Alaska (USA), Iqaluit, Nunavut (Canada), and Thule, Greenland. A point of emphasis of ARCTAS-B was characterizing the effects of biomass burning emissions from the forest fires ubiquitous during the Arctic summer and examining the chemistry within smoke plumes of varying age (Jacob et al., 2010). ARCTAS-B was based in Cold Lake, Alberta (Canada) and Thule, Greenland.





This study is different from previous sensitivity studies involving CTMs, specifically GEOS-Chem, because the quantity and diversity of perturbed inputs are greater. Through a global sensitivity analysis, we identify and quantify the sources of uncertainty for atmospheric oxidants and explore how these factors explain model–measurement differences. Presented here is a global sensitivity analysis of a global CTM allowing for the assessment of model uncertainties and determining the sensitivities of model outputs to chemistry, emissions, and meteorology input factors.

## 2 Methods

### 2.1 GEOS-Chem

The chemical transport model used for this study is GEOS-Chem. GEOS-Chem has been a valuable tool in understanding global air chemistry since its introduction into the literature (Bey et al., 2001) and is currently used by scores of institutions around the world for a wide ranging set of air chemical applications. This study uses the standard GEOS-Chem CTM (v9-02). For computational expediency, the model runs use a regridded horizontal resolution of 4° x 5° and 47 hybrid vertical layers. While previous CTM studies have shown that coarse resolution elevates OH concentrations and ozone production rates, the error from resolution typically pales in comparison to those errors arising from chemistry, meteorology, and emissions (Wild and Prather, 2006). In our case, we found small differences (usually < 10 %) for ARCTAS-A and B between mean vertical profiles of ozone, OH, and $HO_2$ using either 4° x 5° or 2° x 2.5° resolutions and thus using the coarser resolution is adequate for this study. The following sections briefly describe the meteorology, emissions, and chemistry components of the model.

### 2.1.1 Meteorology

GEOS-Chem is driven by the Global Modeling and Assimilation Office's (GMAO) GEOS-5 (Goddard Earth Observing System) meteorological model. GEOS-5 has a native resolution of 0.5° x 0.666° with 72 hybrid eta levels but is regridded to 4° x 5° with 47 hybrid vertical levels for input into GEOS-Chem. There are about 60 GEOS-5 meteorological fields handled by GEOS-Chem. Mixing depths and surface meteorological fields, such as soil wetness, heat fluxes, and albedo have a 3 hour temporal resolution. In contrast, 3D fields, such as u and v wind components and temperature, have 6 hour temporal resolution (Bey et al., 2001). Transport is handled by the semi-Lagrangian TPCORE algorithm (Lin and Rood, 1996).

Due to the lack of published uncertainties associated with the GEOS-5 meteorological data, we defined our meteorological uncertainties as the average monthly standard deviation of the difference between GEOS-5 and GEOS-4 meteorological fields for 2005, a year of overlap between the models. For relative and specific humidity, an uncertainty of 5 %, similar to Heald et al. (2010) was assumed. Cloud mass flux uncertainty was inferred from differences between GEOS-5, a single column model, and a cloud resolving model and set at a factor of 1.5 (Ott et al., 2009).





### 2.1.2 Emissions

GEOS-Chem includes emissions from a variety of anthropogenic, biogenic, and other emissions sources. For this study, the default emissions were generally used. We note in the following section these exceptions and a more detailed description of the various emissions inventories.

For biogenic emissions, this study used the default MEGAN 2.1 (Model of Emissions and Gases and Aerosols from Nature). Out of the 9 species provided by MEGAN, isoprene emissions are dominant, accounting for about half of the biogenic volatile organic compound (VOC) emissions in GEOS-Chem. We assume a factor of 2 uncertainty for isoprene emissions (Guenther et al., 2012). Biomass emissions, a point of emphasis in the ARCTAS-B campaign, were supplied via the Global Fire Emissions Database 3 (GFED-3) (van der Werf et al., 2010). GFED-3 emissions were calculated every three hours. For both biomass and soil $NO_x$ emissions we assume a factor of 3 uncertainty (Jaeglé et al., 2005).

For anthropogenic volatile organic compound (VOC) emissions, the model uses a combination of REanalysis of the TROpospheric chemical composition (RETRO), Emission Database for Global Atmospheric Research (EDGAR), and regional emissions inventories. RETRO was developed by The Netherlands Organization for Applied Research (TNO). GEOS-Chem 9-02 uses 12 VOC species from RETRO (Reinhart and Millet, 2011). EDGAR v4.1 emissions (Olivier et al., 1996) are the default model for $NO_x$ ($NO_x \equiv NO + NO_2$), CO, and $SO_x$ ($SO_x \equiv SO_2 + SO_4^{2-}$) in GEOS-Chem. It has a resolution of $1°$x $1°$and is available on a yearly basis. For many parts of the world, especially the developed world, this study used the default regional emissions datasets that overwrote the RETRO or EDGAR fields.

Lightning $NO_x$ is emitted through the scheme developed by Price and Rind (1992) in which lightning frequency is parameterized based on cloud height and land cover type. In this scheme, continental flash frequencies are higher than marine storms due to stronger storm updrafts observed over land. GEOS-Chem assumes a global total of 6 Tg N $yr^{-1}$ as per Martin et al. (2007) and Sauvage et al. (2007). For this study, the lightning $NO_x$ emissions were rescaled to 6.3 Tg N $yr^{-1}$ with an assumed uncertainty of $\sim$25 % consistent with more recent literature (Miyazaki et al., 2014). This uncertainty may be higher (Liaskos et al., 2015) but is not a major consideration in this domain given the low lightning frequency in the Arctic.

An important factor for any study of ozone is the stratospheric–tropospheric exchange (STE) of ozone. In GEOS-Chem, it is typically parameterized by the Linoz scheme (McLinden et al., 2000). To allow constant scaling of STE ozone, this study used instead the Synoz algorithm, which exchanges 500 TG $yr^{-1}$ of ozone through the tropopause (McLinden et al., 2000). The assumed uncertainty for this STE ozone is a factor of 2.

### 2.1.3 Chemistry

The standard chemical scheme in GEOS-Chem has more than 230 kinetic reactions. This study uses the Sparse-Matrix Vectorized Gear Code (SMVGEAR) chemical solver (Jacobson and Turco, 1994). These rates are updated periodically and are generally supplied by the Jet Propulsion Laboratory (JPL) (Sander et al., 2011), the International Union of Pure and Applied Chemistry (IUPAC) (Atkinson et al., 2007), or other recent literature. Uncertainties for chemical rate coefficients came from JPL (Sander et al., 2011). The standard photolysis scheme has 55 different reactions and uses the FAST-J algorithm (Wild et al.,



2000) to calculate photolysis rates throughout the troposphere. Uncertainties for photolysis rates came from JPL's combined cross sectional and quantum yield uncertainties (Sander et al., 2011).

### 2.1.4 Heterogeneous chemistry

A major point of emphasis in this study is the effect of the treatment of heterogeneous chemistry in the model, especially the

aerosol particle uptake of $HO_2$ (referred to as gamma $HO_2$). Gamma $HO_2$ is defined as the fraction of $HO_2$ consumed per collision with aerosol particles. Until recent work by Mao et al. (2013a) that proposed catalytic reactions involving copper and iron ions in aqueous aerosols, it was assumed aerosol uptake of $HO_2$ would eventually lead to $H_2O_2$ production (e.g., Jacob, 1986). While $H_2O$ formation is a terminal sink for $HO_x$, $H_2O_2$ can be photolyzed and return $HO_x$ radicals back into the atmosphere. GEOS-Chem has had an inconsistent history in the treatment of $HO_2$ aerosol uptake with both the rate and product

of this reaction. Originally GEOS-Chem set $\gamma_{HO_2} = 0.1$ producing $H_2O_2$ (Jacob, 2000) then $HO_2$ uptake was eliminated from the model to better match tropical results (Sauvage et al., 2007) before the later implementation of Thornton et al.'s 2008 mechanism. In the version of the model used in this study, $HO_2$ heterogeneous aerosol uptake is parameterized by $\gamma_{HO_2} = 0.2$ (Jacob, 2000) yielding $H_2O$, a terminal reaction for $HO_2$ (Mao et al., 2013a) . Uncertainties for heterogeneous chemical factors came from JPL (Sander et al., 2011).

## 2.2 Global sensitivity analysis

The global sensitivity analysis method used in this study is the Random Sampling-High Dimensional Model Representation (RS-HDMR) (Li et al., 2001; Rabitz and Ališ, 1999). RS-HDMR is an approach to the HDMR method in which the inputs are randomly sampled from their uncertainty distributions. This study employed a slight variation of the RS-HDMR method in which, in lieu of randomly sampling the input space, it is sampled using a Sobol Sequence (Sobol, 1976), a quasi–random

number sequence. Using this sequence allows for more efficient sampling of the input space and quicker convergence of the RS-HDMR metamodel solution (Feil et al., 2009), an important advantage with the high computational costs associated with chemical transport models. The HDMR method describes the model output as an expansion in terms of the input factors.

$$f(x) = f_0 + \sum_{i=1}^{n} f_i(x_j) + \sum_{1 \leq i \leq n} f_{ij}(x_i, x_j) + ... + f_{12...n}(x_1, ..., x_n) \tag{1}$$

Here $f_0$ is the zeroth order component, a constant equivalent to the mean (Eq. 2), $f_i$ is the first order effect corresponding to

the independent effect of the input $x_i$ on the output (Eq. 3), $f_{ij}$ corresponding to the second order effect on the output of inputs $x_i$ and $x_j$ working cooperatively (Eq. 4), on down to the $n^{th}$ order effect on the output by all the inputs working cooperatively (Rabitz and Ališ, 1999).

$$f_0 \approx \frac{1}{N} \sum_{s=1}^{N} f(x^s) \tag{2}$$

$$f_i \approx \sum_{r=1}^{k_i} \alpha_r^i \varphi_r^i(x_i) \tag{3}$$



$$f_{ij}(x_i, x_j) \approx \sum_{p=1}^{l_i} \sum_{q=1}^{l_j} \beta_{pq}^{ij} \varphi_p^i(x_i) \varphi_q^j(x_j) \tag{4}$$

Here $\varphi$ represents orthonormal polynomials, $k_i$, $l_i$, and $l_j$ represent the orders of the polynomials, $\alpha$ and $\beta$ are constant coefficients.

When using the RS-HDMR approach, the component functions representing the different ordered effects are orthogonal to one another. Because of this property, the total variance can be decomposed into a sum of variances of each component function (e.g., Li et al., 2010; Chen and Brune, 2012). For example:

$$V(f(x)) = \sum_{i=1}^{n} V(f_i(x_i)) + \sum_{1 \leq i \leq n} V(f_{ij}(x_i, x_j)) + ... + V(f_{12...n}(x_1, ..., x_n)) \tag{5}$$

Where $V(f_i(x_i))$ represents the variance of the first order effect due to the input $x_i$ and so forth. It is important to note that $f_i(x_i)$ (Eq. 3) is not necessarily best described by a first order polynomial. From this expansion of the variance, the sensitivity indices of each component can be found by normalizing Eq. (5) by the total variance. Should $\Sigma S_i \approx 1$, first order effects dominate and individual second order effects do not need to be calculated.

$$S_i = \frac{V(f_i(x_i))}{V(f(x)))} \tag{6}$$

$$S_{ij} = \frac{(V(f_{ij}(x_i, x_j))}{(V(f(x)))} \tag{7}$$

Due to the relatively long run time and the large number of inputs that go into the GEOS-Chem model, a Morris Method sensitivity test (Morris, 1991) for the Arctic domain was completed before starting the RS-HDMR study. The Morris Method, also known as the Elementary Effects method, is a computationally inexpensive method to qualitatively determine which model factors have effects that are negligible, linear, or non-linear and has been used in conjunction with many previous HDMR studies (e.g., Ziehn et al., 2009; Chen et al., 2012; Lu et al., 2013). As suggested by Saltelli et al. (2008), we employed 10 trajectories and 4 discrete levels within the uncertainty distributions for sampling. Initially, 465 different model inputs were perturbed. In the name of computational expediency, the number of perturbed inputs was reduced to approximately the 25 % most important factors for the remaining 8 trajectories. As the Morris Method tests were used to prescreen factors for inclusion into the RS-HDMR tests, this initial cull after two trajectories did not influence the factors chosen at the conclusion of the Morris Method test.

After the Morris Method tests were completed, we selected the 50 most influential factors for $HO_2$, OH, and ozone concentrations for the spatial domain corresponding to the ARCTAS mission. This limiting the analysis to 50 factors is in line with (Ziehn and Tomlin, 2008b); however, they note that this pre-screening process may not be necessary if thresholds are implemented in constructing the HDMR metamodel to exclude unimportant factors. In addition to the 50 most influential factors, regional Canadian $NO_x$ emissions from the Criteria Air Contaminant (CAC) inventory, and methyl bromoform emissions were





also included in our HDMR analysis. Methyl bromoform emissions were included in the HDMR tests due to the importance of halogen chemistry in Arctic (e.g., Simpson et al., 2007). All the factors included in the RS-HDMR analysis are listed in Table 1.

### 2.2.1 Uncertainties

After determining the factors to include in the HDMR test, the next step was to create the distributions from which to sample. Uncertainties for all the factors are listed in Table 1. Lognormal distributions were used for all distributions, except those for temperature, soil wetness, relative humidity, and cloud fraction for which normal distributions were used. Standard deviations for the lognormal uncertainty distributions were determined by $\sigma = f - 1$, where f is the published uncertainty factor and $\sigma$ is the standard deviation of the distribution to be sampled, similar to Stewart and Thompson (1996). To ensure $\sim$95 % of the

quasi-random samples would be within the published uncertainty bounds and reflecting the $2\sigma$ range JPL uses to incorporate chemical kinetic data and inferred from emissions uncertainties, these standard deviations were then halved before creating the distributions.

With the uncertainty distributions created, a Sobol Sequence (discarding the first 512 sets of values as spin up) was created to quasi-randomly sample these distributions and perturb the model. To ensure model perturbations had time to spread and

reach a new global equilibrium, a 9 month spin-up period was employed before the first flights in April 2008. The ensemble was limited to 512 model runs. While previous implementations of the RS-HDMR to box models used thousands of runs (e.g., Chen and Brune, 2012), recent use of the method with a land surface model shows reliable results with as few as 256 runs (Lu et al., 2013). Likewise, we found little difference in results between 512 and 256 model runs, but have included all 512 in this study.

### 2.2.2 Calculation of sensitivity indices

Graphical User Interface-HDMR (GUI-HDMR) was used to calculate all the sensitivity measures and analyze the input–output behavior of the model (Ziehn and Tomlin, 2009). This MATLAB software package is freely available through http://www.gui-hdmr.de. For use within the software, the values of the inputs were rescaled according to their respective percentiles within the uncertainty distributions. We employed the correlation method provided in the GUI-HDMR software (Kalos and Whitlock,

1986; Li et al., 2003), a variance reduction method. In using the correlation method, the construction of the RS-HDMR expansion becomes an iterative process using an analytical reference function. With this method, as noted in (Li et al., 2003), the accuracy of the RS-HDMR expansion increases without a corresponding increase in ensemble size, a valuable advantage considering the expensive nature of running CTMs.

### 2.3 Measurements

For comparison to the model, we also used measurements collected aboard the NASA DC8 airplane. OH and $HO_2$ measurements came from Pennsylvania State University's Airborne Tropospheric Hydrogen Oxides Sensor (ATHOS) (Faloona et al.,





2004). ATHOS uses Laser Induced Fluorescence (LIF) to measure $HO_x$ mixing ratios. The National Center for Atmospheric Research's (USA) (NCAR) Selected-Ion Chemical Ionization Mass Specrometer (SI-CIMS) and Peroxy Radical Chemical Ionization Mass Spectrometer (PeRCIMS) also measured OH and $HO_2$ respectively aboard the DC8. Comparisons between the methods showed good agreement during the campaign (Ren et al., 2012). For the purposes of our analysis, only ATHOS

measurements are considered. Ozone observations aboard the DC8 were measured by NCAR using the chemiluminescence method (Weinheimer et al., 1994).

Since ARCTAS, interferences have been found in the measurements of both OH (Mao et al., 2012) and $HO_2$ (Fuchs et al., 2011). The OH interference can be anywhere from 20 % to 300 % of the actual ambient OH, while the $HO_2$ interference is typically less than a factor of two. Both interferences require the presence of alkenes or aromatics and so are limited to planetary

boundary layer environments in which these volatile organic compounds are common. Interferences in the free troposphere and over much of the Arctic will be negligible.

## 2.4 Data manipulation

To compare aircraft observations to the model ensemble, the Planeflight option within GEOS-Chem was used. The Planeflight option allows for modeled values to be output at one minute intervals along the DC8 flight track. To match the modeled flight

track, we averaged the aircraft observation data over one minute intervals and excluded observations from the stratosphere. For our flight-by-flight HDMR analyses, average mixing ratios along the flight track as the output of interest in GUI-HDMR were used. For vertical profiles, modeled and measured flight track data were binned and averaged in one kilometer increments, excluding the transit flights (flights 3, 11, 16, and 24). While it is a concern that the modeled representation of the flight tracks may misrepresent spatially or temporally synoptic or mesoscale features important to the abundances of the studied species,

these differences likely are small when averaged over each flight, and especially when averaged across all modeled flights. At this time, Planeflight offers the most consistent method for model–measurement comparison.

## 3 Results

Given the seasonal differences between Arctic spring and summer in both meteorology and emissions, and the differences between the mission objectives between ARCTAS-A and ARCTAS-B, the results are separated by their respective season.

During both ARCTAS-A and ARCTAS-B, the NASA DC8 sampled the troposphere at a variety of heights ranging from near surface to the lower reaches of the stratosphere providing a representative view of the Arctic troposphere as seen in the bar graphs in Figs. 2 and 6.



### 3.1 ARCTAS-A (Spring 2008)

#### 3.1.1 Uncertainty analysis

Across the modeled ensemble, ozone has relatively low uncertainty (6.8 %, $1\sigma$ confidence) reflecting the low ozone production rates within the domain during ARCTAS-A (Fig. 1). In contrast to ozone, we found both OH and $HO_2$ to have much higher
uncertainty across the model ensemble with OH and $HO_2$ both having $1\sigma$ uncertainties of around 27 %. Figure 2 shows this uncertainty spread vertically. For ARCTAS-A, uncertainties and sensitivities were generally uniform with altitude across the model ensemble for ozone and $HO_x$.

#### 3.1.2 Vertical profiles

Figure 2 shows mean vertical profiles binned per kilometer for the spring deployment (Fig. 1). Ozone was consistently un-
derpredicted by the model at all altitudes except near the surface and showed little variation across the ensemble in modeled ozone. The lack of significant in situ ozone production in April over the domain could partially explain the small variation in modeled mixing ratios among ensemble members. Similar to Mao et al. (2010), OH mixing ratios were low, in the tenths of one ppb and showed a consistent model underestimation for the lower and middle troposphere with better agreement above $\sim$6 km, although the limit of detection for the OH measurement is $\sim 10^5$ cm$^{-3}$. Across the model ensemble there is general
agreement between measured and modeled $HO_2$ within the vertical column as measured values are mostly within the first standard deviation of modeled results. This is different from Mao et al. (2010) in which GEOS-Chem showed a consistent overestimation of $HO_2$. Above 7 km, modeled $HO_2$ is higher than measured, by upwards of a factor of 2, similar to Mao et al. (2010). These results are consistent with improvement in modeled characterization of $HO_2$ aerosol particle uptake as aerosol concentrations are highest in the lowest few kilometers of the atmosphere and very low in the upper reaches of the troposphere.

#### 3.1.3 Sensitivity analysis

Figure 3 shows the first order results of the HDMR analysis for the average tropospheric mixing ratios along selected flight tracks for ozone, OH, and $HO_2$. For $HO_x$ and ozone, the sensitivities are, with a minor few exceptions, altitude independent. The first order sensitivity index for all factors are represented and are color coded by their respective category as defined in Table 1. In this sense, first order effects describe each factor's individual contribution to the ensemble variance. The RS-HDMR
component functions for each factor are not necessarily linear, and are in fact often best represented by $2^{nd}$ degree and higher polynomials. GUI-HDMR calculates the optimal order for each HDMR polynomial using a least squares method (Ziehn and Tomlin, 2008a). The missing portion of the pie graph represents second and higher order sensitivities. While all flights are not presented here, the three flights in Fig. 3 cover the geographic spread of the domain and are representative of the results seen among other spring flights.

Ozone: Overall, the sum of all the first order effects was usually below 0.90 meaning that first order effects explain close to 90 % of the observed variance. To calculate meaningful second order terms will require substantially more model runs.



For each spring flight the photolysis of $NO_2$ was the most influential factor for modeled ozone with sensitivity indices ranging from around 0.09 to 0.11 (mean 0.10). It is not surprising $NO_2$ photolysis is a sensitive factor considering the photolysis of $NO_2$ leads directly to ozone production; however, it is somewhat surprising given its rather low uncertainty (20 %) and the limited ozone production in the Arctic spring. Other most influential factors are the $NO_2$ + OH reaction (mean $S_i$ = 0.083),

soil $NO_x$ emissions (0.047), temperature (0.056), and methyl bromoform emissions (0.072). Sensitivity of ozone to methyl bromoform emissions is expected due to bromine compounds' ability to catalytically destroy ozone, especially early in the Arctic spring when sunlight returns allowing for halogen photochemistry to commence (e.g., Barrie et al., 1988). Tropospheric ozone depletion events arising via catalytically destructive halogen reactions were observed during the ARCTAS-A campaign, mainly below 1 km (Koo et al., 2012).

OH: OH mixing ratios were very low, in the tenths of one ppb. These low mixing ratios are expected considering the low sun angles in April over the Arctic and was noted in prior ARCTAS studies (Mao et al., 2010). Unlike ozone, $\sum S_i \approx 0.90$ for most modeled flights meaning first order effects describe the vast majority of the model uncertainty. For all the flights, aerosol particle uptake of $HO_2$ (gamma $HO_2$) was the most influential factor having $S_i$ values ranging from 0.37 and 0.58 (mean $S_i$ = 0.49). Temperature (0.071), biomass CO (0.058) also routinely had $S_i$ values above 0.05. Among emissions, Asian

and biomass $NO_x$ and CO contributed the most to the uncertainty. The influence of Asian emissions during ARCTAS-A has been noted previously (Jacob et al., 2010) and highlights the sensitivity of the Arctic region to the advection of anthropogenic pollution.

$HO_2$: As with OH, $HO_2$ mixing ratios were also low, and first order effects dominated in the RS-HDMR metamodel with $\sum S_i$ values ranging from 0.94 to 0.98. Of the first order effects, gamma $HO_2$ was dominant, with $S_i$ values ranging from 0.60

to 0.76 (mean $S_i$ = 0.71). This suggests that around 71 % of the uncertainty associated with modeled $HO_2$ is due to uncertainties in gamma $HO_2$. Temperature was the only other factor regularly having a sensitivity index greater than 0.05 (mean $S_i$ = 0.10).

Aerosol particle uptake of $HO_2$ has been found in previous studies to be of particular importance in the Arctic (Martin et al., 2003; Mao et al., 2010). With low $NO_x$ concentrations and temperatures, the $HO_2$ lifetime in the Arctic spring is especially long when compared to the midlatitudes or tropics. Without terminating reactions with other $NO_x$ or $HO_x$ radicals, uptake by

aerosols becomes a dominant loss of $HO_2$.

Providing a broad view of the sensitivity results from ARCTAS-A, Fig. 4 shows the same analysis as Fig. 3 but averaged across all flights and summed by factor category as defined in Table 1. While ozone is most sensitive to emissions, chemical factors from kinetics and photolysis rates also contribute a large portion to the uncertainty. OH and $HO_2$ are overwhelmingly sensitive to heterogeneous chemistry, particularly gamma $HO_2$ as seen in Fig. 3.

**3.2 ARCTAS-B (Summer 2008)**

### 3.2.1 Uncertainty analysis

Compared to ARCTAS-A, ozone in ARCTAS-B (Fig. 5) saw much higher uncertainty across the model ensemble (12 %, $1\sigma$ confidence) compared to the spring (6.8 %). This is reflective of the more photochemically active summertime in contrast to





the spring. Like the spring, OH and $HO_2$ uncertainties were similar to the spring with OH and $HO_2$ uncertainties being 25 % and 24 % ($1\sigma$ confidence) respectively across the model ensemble.

### 3.2.2 Vertical profiles

Figure 6 shows the vertical profiles observed in ARCTAS-B for ozone, OH, and $HO_2$. As also reported by Alvarado et al.

(2010), we found GEOS-Chem to under-predict ozone for the middle troposphere by 10–20 ppb. OH mixing ratios, as in ARCTAS-A, were low. Although well predicted by the model above 3 km, OH was over-predicted below 3 km by around a factor of 2. $HO_2$ saw the greatest model–measurement disagreement with the model under-predicting $HO_2$ by over a factor of 2 below 2 km. This modeled underestimation of $HO_2$ is noteworthy considering $HO_2$ overestimation is much more common in air chemistry models (e.g., Mao et al., 2013a). Even when excluding measurements taken within smoke plumes as defined

by HCN > 1000 pptv, this underestimation decreases only by about 1 pptv for the lower 2 km and remains about a factor of 2. The simultaneous overestimate of OH and underestimate $HO_2$ suggests the model is partitioning $HO_x$ incorrectly and may be missing or underrepresenting OH reactions that would cycle OH to $HO_2$. Another possible explanation for a portion of this overestimation of $HO_2$ could be organic peroxy radical ($RO_2$) interference artificially elevating $HO_2$ measurements (Fuchs et al., 2011), but this would likely not account for the factor of 2 underestimation.

### 3.2.3 Sensitivity analysis

First order RS-HDMR sensitivity indices for tropospheric average ozone, OH, and $HO_2$ for along the path of flights 17, 19, and 22 (Fig. 5) is shown in Figure 7. Figure 8 provides a broad view of the sensitivities calculated across all the ARCTAS-B flights binned by category as shown in Table 1. With a few exceptions, $\sum S_i \approx 0.90$ for all flight averaged ozone, OH, and $HO_2$ meaning first order effects explain around 90 % of the model uncertainty with higher order input interactions responsible for

the remaining uncertainty. Compared to ARCTAS-A, emissions are more influential across the board, especially from soils, biomass, and isoprene. Like ARCTAS-A, ARCTAS-B sensitivities were largely altitude independent.

Ozone: For modeled ozone, mixing ratios were most sensitive to soil $NO_x$ emissions with average $S_i$ across the flights around 0.181, isoprene emissions (mean $S_i = 0.081$), biomass CO and $NO_x$ emissions (mean $S_i = 0.069$, 0.089 respectively), the $NO_2$ + OH reaction rate (mean $S_i = 0.075$), and $NO_2$ photolysis (mean $S_i = 0.054$). The greater sensitivity to emissions

in the summer compared to spring is almost certainly a result of biomass, soil, and isoprene emissions being much greater in Arctic summer than spring. These higher emissions coupled with higher sun angles allows for ozone production in the Arctic summer, unlike the very slow production in spring. Also, there is relatively low sensitivity to anthropogenic emissions, reflecting the remoteness of this domain and its relative pristine condition.

OH: Soil and biomass $NO_x$ emissions (mean $S_i$ across flights is 0.095 and 0.105 respectively), biomass CO emissions (mean

$S_i = 0.220$), and gamma $HO_2$ (mean $S_i = 0.137$) are most influential for OH. As normal OH production requires the photolysis of ozone, OH being sensitive to the same emissions as ozone is expected. OH is sensitive to gamma $HO_2$ as it represents a net sink of $HO_x$ radicals.





$HO_2$: For $HO_2$, the modeled mixing ratios were most sensitive to gamma $HO_2$ and biomass CO and organic carbon emissions with mean $S_i$ across the flights of 0.405, 0.167, and 0.094 respectively. This is qualitatively similar to the results from the spring, only the dominance of gamma $HO_2$ on the total variance in modeled $HO_2$ is lessened, but still prominent (mean $S_i$ = 0.405 in summer as opposed to 0.712 in spring). It is noteworthy that even with reduced $HO_2$ lifetimes in the Arctic summer compared to spring, $HO_2$ still had such high sensitivity to gamma $HO_2$.

Fig. 8 shows an overview of the sensitivity results from ARCTAS-B averaged among all flights and summed by factor category as defined in Table 1. As found during ARCTAS-A (Fig. 4), ozone is most sensitive to emissions with chemical factors from kinetics and photolysis rates also contributing a large portion of the uncertainty. In contrast to the spring, OH and $HO_2$ are most sensitive to emissions factors in the summer; however, heterogeneous chemistry, especially gamma $HO_2$, provides a large slice of the uncertainty as also noted in the spring (Fig. 4). In the case of summer $HO_2$, gamma $HO_2$ contributes individually almost as much as the sum of all emissions factors to the model uncertainty.

To probe this disagreement between modeled and measured $HO_2$ at lower altitudes seen in Fig. 6, we examined ensemble members with the best agreement between modeled and measured $HO_2$ profiles. The ensemble members that matched the measured profile best had especially low gamma $HO_2$ values. Figure 9 shows a comparison between the entire ensemble and ensemble members with gamma $HO_2$ values in the lowest 10 percentile of the uncertainty distribution ($\gamma_{HO_2}$ < 0.04). This model–measurement disagreement was not observed among all flights in the ARCTAS-B campaign. In fact, areas with lower aerosol abundances such as the northernmost flights, 22 and 23, showed general agreement between modeled and measured $HO_2$ profiles (Fig. 10). Likewise, above 4 km, the model performs very well in replicating the observed $HO_2$ profile. Given its overwhelming importance in the RS-HDMR analysis, mischaracterization of gamma $HO_2$ is a likely cause.

One possible cause of this disagreement is that $HO_2$ aerosol particle uptake is leading to the formation of $H_2O_2$ instead of $H_2O$. Figure 11 shows the modeled and measured $H_2O_2$ profile for the ARCTAS-A and B flights. When altering the model for gamma $HO_2$ to produce $H_2O_2$ instead of $H_2O$ ($\gamma_{HO_2} \Rightarrow 0.5\ H_2O_2$) (blue lines in vertical profiles in Figs. 2, 6, and 11), modeled $HO_2$ increased throughout the vertical column by between 0.25 and 0.75 ppt in the summer (Fig. 6) and between 0.5 and 1 ppt in the spring (Fig. 2). In this same model run, $H_2O_2$ increased upwards of a factor of 3, especially in the lowest 2 km taking modeled values a factor of 2 or greater higher than measurements (Fig. 11). It is noted that there was a large spread in $H_2O_2$ within the ensemble and a large uncertainty in the measured values (50 % + 150 ppbv). While the difference in modeled $HO_2$ between model runs having gamma $HO_2$'s product being either $H_2O$ or $H_2O_2$ is important during the spring when $HO_2$ mixing ratios are lower, as Mao et al. (2010) and Figure 2 show, this difference is less significant during the summer when $HO_2$ concentrations are higher (Fig. 6). The difference between these model scenarios cannot be responsible for the difference between the observed and modeled mixing ratios in the lowest 2 km ($\sim$7 to 8 ppt). This small effect suggests that, especially in the Arctic summer, concentrating on better characterization of the rate may be more important than the product for improving the agreement between measured and modeled $HO_x$.





## 4   Conclusions

We have applied a RS-HDMR sensitivity analysis to a 3D chemical transport model. First order sensitivity indices for the 52 perturbed model inputs have been calculated and shown in Figs. 3, 4, 7, and 8. For OH and $HO_2$, we find general agreement between modeled and measured values when uncertainties in the measurements and uncertainties in model input factors are

taken into account as evidenced by the overlap between the vertical model and measurement profiles (Figs. 2 and 6) with the notable exception of summertime $HO_2$. In contrast, vertically binned modeled and measured ozone mixing ratios do not show as much overlap, especially in spring. Mischaracterization of advection from the midlatitudes as posited by Alvarado et al. (2010) is a possible source of this error, especially given the importance of isoprene and Asian and North American anthropogenic emissions in the Arctic spring. Other possible sources of error may come from mischaracterized chemistry or

under-represented stratospheric transport. Modeled ozone was most sensitive to various emissions sources, especially soil $NO_x$ and isoprene, and chemical factors, such as j[$NO_2$] and k[$NO_2$]+[OH]. Model sensitivities for OH and $HO_2$ were dominated by aerosol particle uptake of $HO_2$, especially in the spring with a combination of biomass and soil emissions being also important, particularly in summer. While the sensitivity of oxidants to emissions is expected considering the high uncertainty in emissions inventories (factors of 2 to 3), it is noteworthy that chemical kinetic and photolysis rates also were responsible

for a considerable portion of uncertainty even with their much lower published uncertainties, 20 % and 30 % for j[$NO_2$] and k[$NO_2$]+[OH] respectively for example. This highlights the value in not only more certain emissions inventories but also more certain chemical kinetics rates.

HO$_2$ aerosol particle uptake remains the dominant source of uncertainty in our analysis for $HO_x$. From our ensemble, the best model–measurement agreement came with lower gamma $HO_2$ values ($\gamma_{HO_2} < 0.04$) than currently implemented in

GEOS-Chem regardless of the uptake product. Much attention has been given to determining the product of the aerosol particle uptake of $HO_2$, and whether or not or in which instances $H_2O_2$ or $H_2O$ is produced. We find there is not a large difference in modeled $HO_2$ between these two possibilities, especially in Arctic summer. In contrast, $H_2O_2$ is very sensitive to the product of the aerosol particle uptake of $HO_2$ with $H_2O_2$ increasing upwards of a factor of 3 when the product is $H_2O_2$ instead of $H_2O$ (Fig. 11). Recent studies have expanded this question of $HO_2$ uptake products from aqueous aerosols to smaller cloud

droplets (Whalley et al., 2015). In particular, the analysis of Whalley et al. showed the Arctic region being especially sensitive to changes in $HO_2$ uptake compared to the midlatitudes and tropics due to longer $HO_2$ lifetimes in the Arctic. As shown in our results, this study also finds the Arctic region particularly sensitive to gamma $HO_2$. Because the Arctic is unique in its relatively low $HO_x$ mixing ratios and long $HO_x$ lifetimes compared to the midlatitudes and tropics, future research will be needed to determine whether or not gamma $HO_2$ is as important globally as it is in the Arctic and whether or not aerosol particle uptake

rates need to be reduced in GEOS-Chem.

*Acknowledgements.* We could like to acknowledge NASA's Atmospheric Composition Campaign Data Analysis and Modeling program (ACCDAM) for funding this project (grant NNX14AP43G), Harvard University for managing and supporting GEOS-Chem, GEOS-Chem



support for assistance, Andrew Weinheimer of NCAR for ozone measurements, and Paul Wennberg and the CalTech group for $H_2O_2$ measurements.



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



**Table 1.** Factors included in RS-HDMR analysis and their respective uncertainties. OC is organic carbon, ALK$_4$ is lumped $\geq$4C alkanes, MP is methylhydroperoxide, and MO$_2$ is methylperoxy radical. Uncertainties are expressed as multiplicative factors, except as noted in meteorological factors.

| Factor | Uncertainty# | Factor | Uncertainty# |
|---|---|---|---|
| **Emissions** | | **Photolysis** | |
| Biomass CO, NH$_3$, NO$_x$, OC | 3.0$^a$ | j [BrNO$_3$] | 1.4$^d$ |
| Soil NO$_x$ | | j [BrO] | 1.4$^d$ |
| CAC (Canada) NO$_x$ | | j [H$_2$O$_2$] | 1.3$^d$ |
| Methyl Bromoform (CHBr$_3$) | | j [HNO$_3$] | 1.3$^d$ |
| EDGAR NO$_x$ | | j [HOBr] | 2.0$^d$ |
| EMEP (European) NO$_x$ | | j [MP] | 1.5$^d$ |
| EPA (USA) ALK$_4$, CO, NO$_x$ | 2.0 | j [NO$_2$] | 1.2$^d$ |
| Streets (E. Asian) CO, NH$_3$, NO$_x$, SO$_2$ | | j [O$_3$] | 1.2$^d$ |
| Ship NO$_x$ | | **Meteorology** | |
| Strat-Trop Exchange O$_3$ | | Cloud fraction | 8.5 %$^e$ |
| Isoprene | 2.0$^b$ | Cloud mass flux | 1.5$^f$ |
| Lightning NO$_x$ | 1.25$^c$ | Relative Humidity | 5 %$^g$ |
| **Kinetics** | | Soil Wetness | 8.8 %$^e$ |
| k [BrO] [HO$_2$] | 1.15 / 1.2*$^d$ | Specific Humidity | 5 %$^g$ |
| k [BrO] [NO$_2$] | 1.2$^d$ | Temperature | 1.8K$^e$ |
| k [HNO$_3$] [OH] | 1.2$^d$ | **Heterogeneous** | |
| k [HO$_2$] [HO$_2$] | 1.15 / 1.2*$^d$ | Gamma HO$_2$ | 3.0$^d$ |
| k [HO$_2$] [NO] | 1.15$^d$ | Gamma HOBr | 3.0$^d$ |
| k [MO$_2$] [HO$_2$] | 1.3$^d$ | Gamma N$_2$O$_5$ | 1.4$^d$ |
| k [MP] [OH] | 1.4$^d$ | Gamma NO$_2$ | 3.0$^d$ |
| k [NO$_2$] [OH] | 1.3$^d$ | Henry's Law HOBr | 10.0$^d$ |
| k [O$_3$] [HO$_2$] | 1.15$^d$ | | |
| k [O$_3$] [NO] | 1.1$^d$ | | |
| k [O$_3$] [NO$_2$] | 1.15$^d$ | | |
| k [OH] [CH$_4$] | 1.1$^d$ | | |

# at 1$\sigma$ uncertainty confidence; *high pressure limit / low pressure limit uncertainties; $^a$Jaeglé et al. (2005); $^b$Guenther et al. (2012); $^c$Miyazaki et al. (2014); $^d$Sander et al. (2011); $^e$GEOS5-GEOS4; $^f$Ott et al. (2009); $^g$Heald et al. (2010)



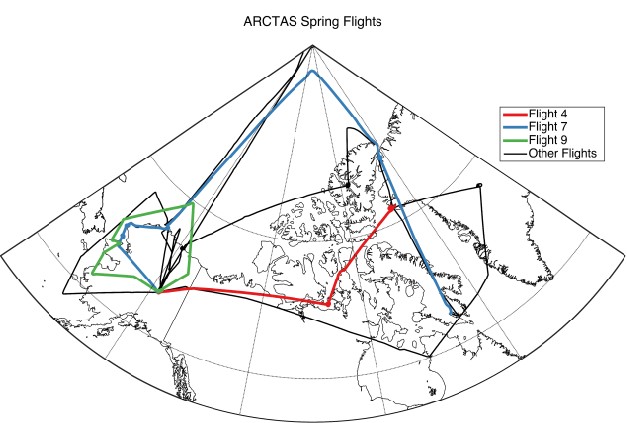

**Figure 1.** Map of ARCTAS-A flights over the North American Arctic. Highlighted flights correspond to flight data results analyzed in Fig. 3

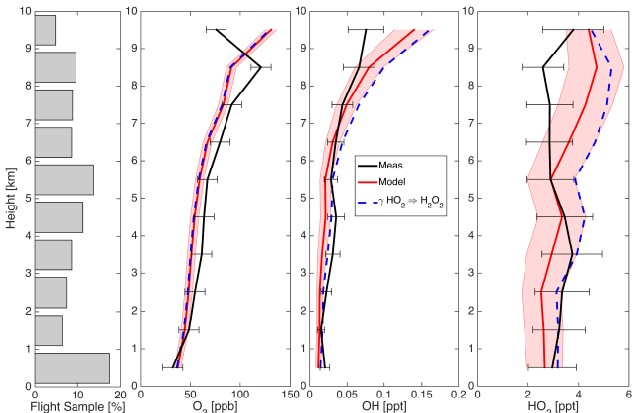

**Figure 2.** Vertical profiles of mean modeled (red) and measured (black) ozone, OH, and $HO_2$ for ARCTAS-A flight data binned by kilometer. Gray bar graph shows percent of flight data within each vertical bin. Shaded regions represent $1\sigma$ of model ensemble; error bars on measurements are uncertainty at $1\sigma$ confidence. Blue line represents gamma $HO_2$ producing $H_2O_2$ rather than $H_2O$.





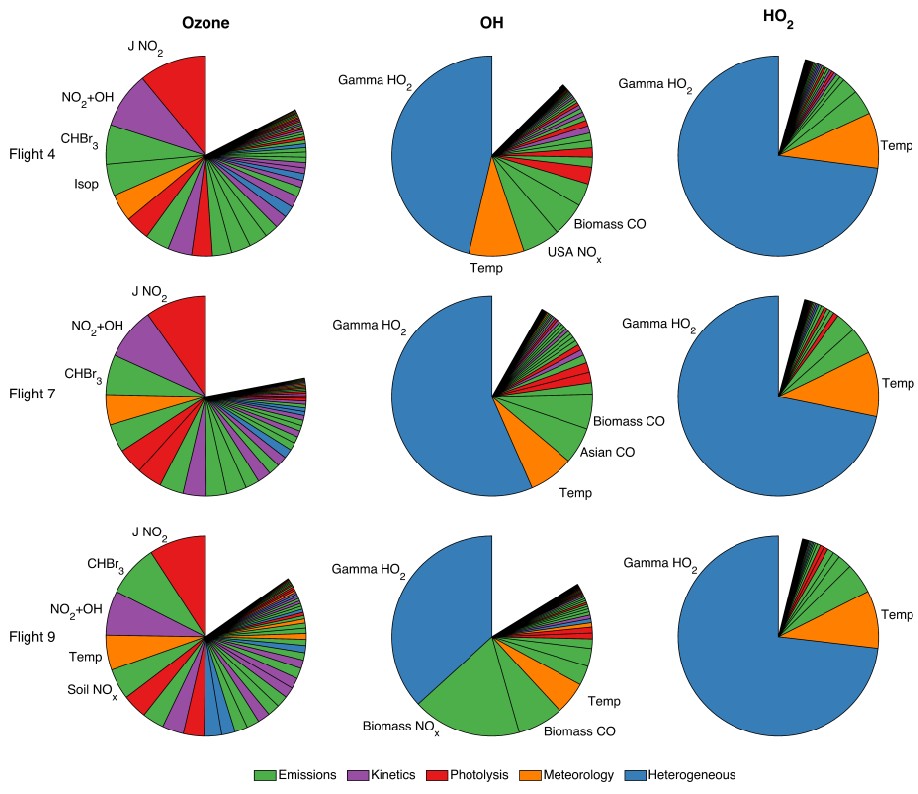

**Figure 3.** First order sensitivity indices for average flight track $O_3$, OH, and $HO_2$ for ARCTAS-A flights. Legend categories are defined in Table 1.

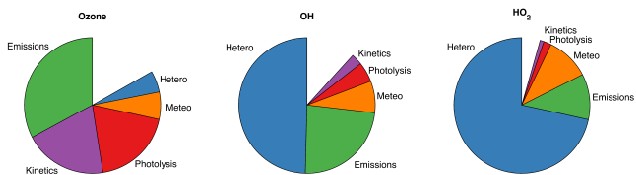

**Figure 4.** First order sensitivity indices for modeled $O_3$, OH, and $HO_2$ during ARCTAS-A averaged across all flights and binned by categories defined in Table 1.




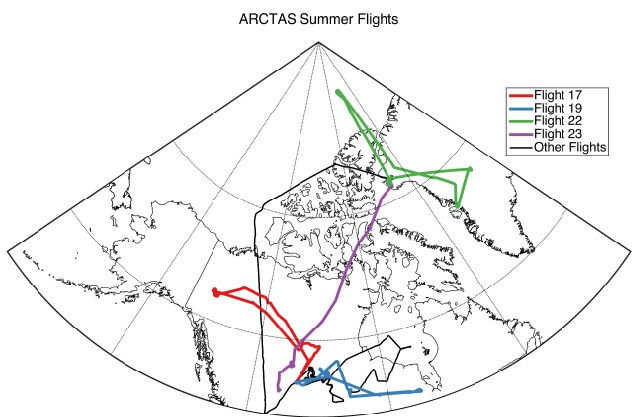

**Figure 5.** Map of ARCTAS-B deployment over the North American Arctic. Colored flights correspond to flight data results analyzed in Figs. 7 and 10.

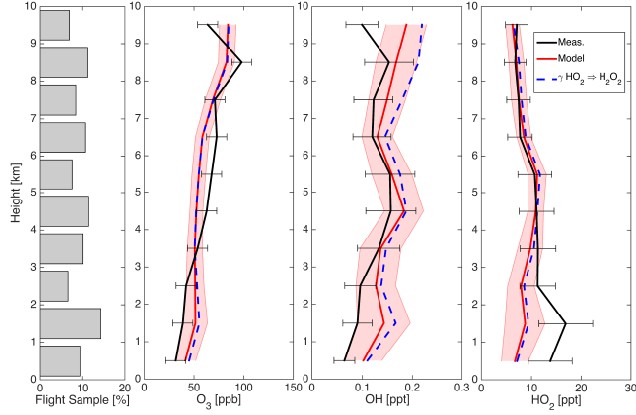

**Figure 6.** Vertical profiles of mean modeled (red) and measured (black) ozone, OH, and $HO_2$ for ARCTAS-B flight data binned by kilometer. Gray bar graph shows percent of flight data within each vertical bin. Shaded regions represent $1\sigma$ of model ensemble; error bars on measurements are uncertainty at $1\sigma$ confidence. Blue line represents gamma $HO_2$ producing $H_2O_2$ rather than $H_2O$.





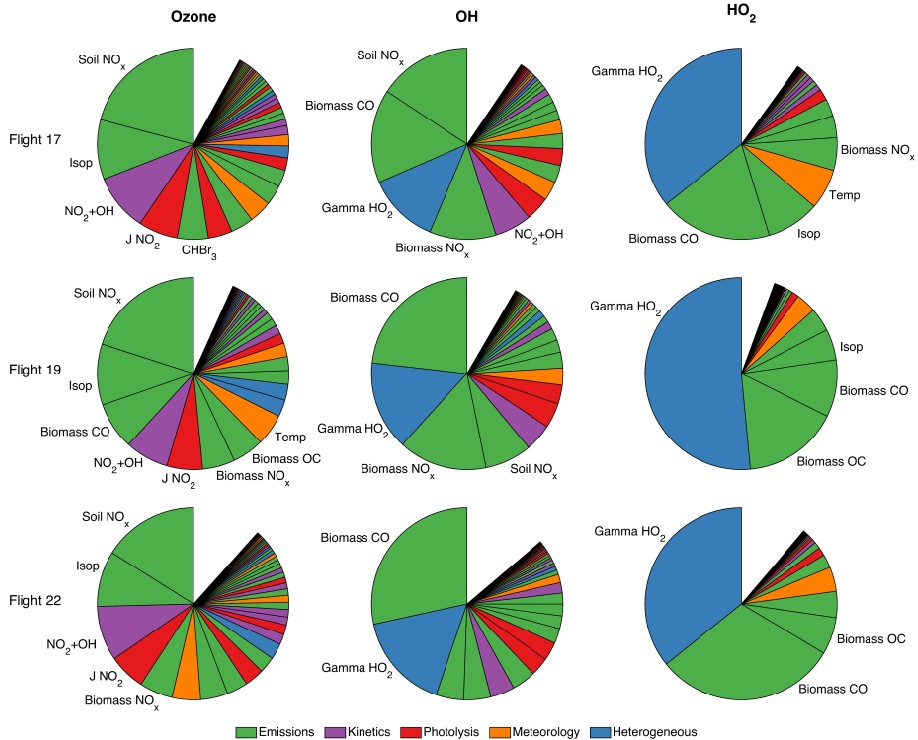

**Figure 7.** First order sensitivity indices for average modeled $O_3$, OH, and $HO_2$ along selected ARCTAS-B flights. Legend categories are defined in Table 1.

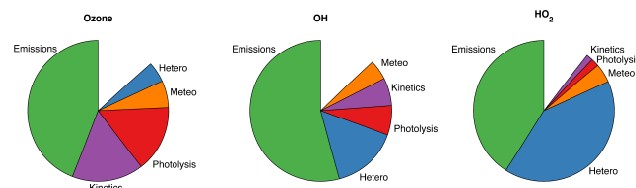

**Figure 8.** First order sensitivity indices for modeled $O_3$, OH, and $HO_2$ during ARCTAS-B averaged across all flights and binned by categories defined in Table 1.




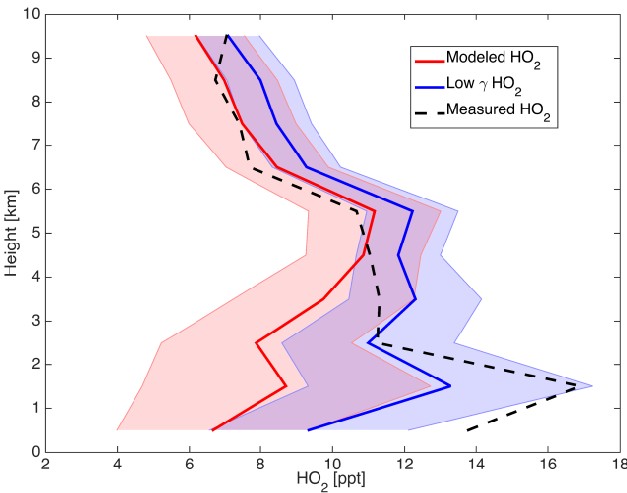

**Figure 9.** Vertical HO₂ profile for ARCTAS-B flights. Shaded region represents $1\sigma$ of the model ensemble. Blue line and region represents model runs with gamma HO₂ values in the lowest 10 % of the uncertainty distribution.

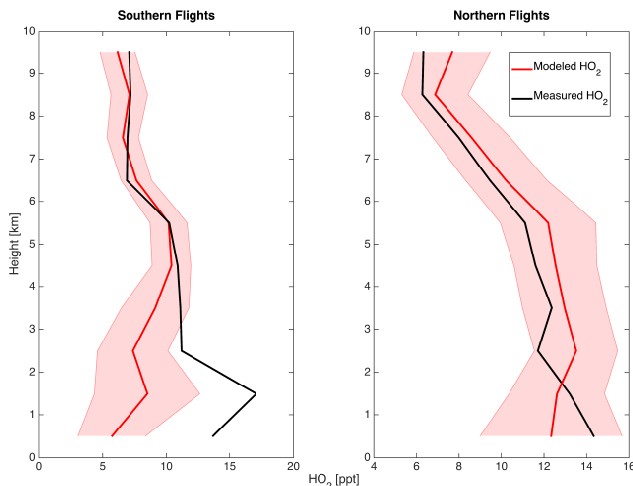

**Figure 10.** Modeled and measured HO₂ profiles for ARCTAS-B flights. Shaded region represents $1\sigma$ of model ensemble. Left represents flights 17, 18, 19, 20, 21. Right represents flights 22 and 23.





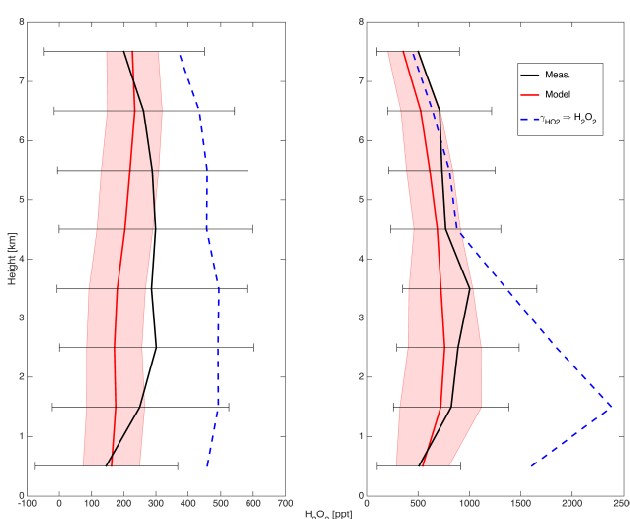

**Figure 11.** Vertical profile for $H_2O_2$ for flights during ARCTAS-A (left) and ARCTAS-B (right). Shaded region represents $1\sigma$ of model ensemble. Error bars represent measurement uncertainty. Blue lines show gamma $HO_2$ producing $H_2O_2$ rather than $H_2O$ in the model.