# Peer review of "Global sensitivity analysis of the GEOS-Chem chemical transport model: Ozone and hydrogen oxides during ARCTAS (2008)"

_Atmospheric Chemistry and Physics, 2016_

## Referee Comment (RC1) · Anonymous Referee #1 · 13 Dec 2016

This paper describes a global sensitivity analysis of Arctic tropospheric oxidants (OH, HO2, ozone) in an ensemble of GEOS-Chem global chemical transport model simulations to uncertainties in emissions, kinetic parameters, and meteorological parameters. Analysis of model response to 50 different parameter uncertainties is presented, and the chief drivers of model uncertainty in ozone and HOx concentrations sampled along DC-8 flight tracks from the ARCTAS 2008 experiments are discussed. The paper identifies a handful of key emission sources, gas-phase kinetic rates, and heterogeneous processes responsible for driving the majority of model ensemble variance. In particular, uncertainty in the HO2 aerosol uptake coefficient is identified as a dominant driver of model uncertainty in HOx, particularly in spring. The paper is well written,

well structured, and has clear figures. The material is a suitable topic for ACP, and will be of wide general interest to those concerned with Arctic atmospheric chemistry and climate, as well as those interested in drivers of model uncertainty in tropospheric chemistry. While the conclusions are interesting and novel, I would recommend that a number of points are addressed before considering accepting the article for publication in ACP.

General comments

1) It would be useful to discuss how model-dependent the large ozone sensitivity to uncertainty in the NO2 + OH reaction rate may be. Previous studies comparing GEOS-Chem with ARCTAS observations have shown that the model displays a large over-prediction of HNO3 and a large under-prediction of PAN in the Arctic troposphere (Figs. 18 & 16 Emmons et al., (2015); Figs. 3 and 4 Arnold et al., 2015). To what extent is the sensitivity to the HNO3 production rate a reflection of the propensity for GEOS-Chem to produce large amounts of HNO3? i.e. is the sink for NO2 through formation of HNO3 (and therefore sensitivity to uncertainty in its rate) realistic? Does this version of the model include the NOy chemistry updates from Fischer et al., (2014) which greatly improved the simulation of NOy chemistry in GEOS-Chem? The authors should include some reference to these past studies comparing GEOS-Chem with ARCTAS data and other models in the discussion, and comment on how the model Arctic NOy budget compares with observations and implications for the inferred sensitivity to the kinetic uncertainties.

2) The large response to soil NOx emissions is a surprising and novel result, and also warrants further discussion. Given the high vertical stability of the Arctic troposphere, there is strong isolation of the free mid / upper troposphere from emissions and processes in high latitude / Arctic boundary layer, and air tends to be transported into the mid/upper troposphere from lower latitudes (e.g. Stohl, 2006, Wespes et al., 2012). Therefore ozone sensitivity at altitudes in the mid and upper troposphere is presumably driven by response to uncertainty in soil NOx emissions from lower latitudes, and

its impacts on ozone chemistry during uplift and long-range transport into the Arctic? It would be useful to expand on this in the manuscript, such that the reader has a better idea of what drives this sensitivity. A factor 3 uncertainty is assumed for these emissions based on Jaegle et al., (2005). Is this the most appropriate and recent reference for framing this uncertainty? Given the importance of this uncertainty for ozone in the N American Arctic, it would be helpful to discuss more widely estimates of the reliability (uncertainty in) the soil emissions if other studies are available and how robust the factor 3 estimate may be.

3) To what extent is the large HOx response to gamma_HO2 a reflection of the large uncertainty range implemented (factor 3)? It would be useful to show what actual range of gamma values this corresponds to. The authors show that the ensemble members with lower gamma values best match profile observations of HO2. How do these gamma_HO2 values compare with those used in previous GEOS-Chem studies? What are the implications for model comparisons with high latitude CO values, which in previous studies have been improved by implementing different formulations of aerosol uptake of HO2 (e.g. Mao et al., 2013)? How does the choice of product ($H_2O_2$ or $H_2O$) affect comparisons with CO and ozone? It would be useful to discuss this, since underestimation of CO at high latitudes in CTMs is a persistent problem (e.g. Emmons et al., 2015).

4) It should be made clear in the abstract and the methodology that this analysis only provides information on drivers of model response to uncertainties in air masses sampled during ARCTAS. It cannot be assumed that this is representative of the whole Arctic unless this can be shown explicitly. Figure 2 shows a good spread of aircraft observations across altitudes, but the flights still only sample the N American Arctic on specific days, when there are certain specific air mass origins.

Specific / minor comments

Page 1, Line 1: "oxidation capability" change to "oxidation capacity"

[Figure]

Page 1, Line 19/20: "Increasing oil and gas exploration and extraction, coupled with summertime shipping lanes through the region will make air pollution worse". This statement needs a reference.

Page 2, line 5: ".. model shortcomings are usually attributed to errors in the chemical reaction rates, emissions, or meteorology (e.g., Wild and Prather, 2006)". The cited study is specifically about effects of model resolution? Please cite examples to back up the specific reasons you list.

Page 2, Line 10: Omit semi-colon.

Page 2, line 13: "two more input factors" should be "two or more input factors"?

Page 4, line 3: Better phrased as: "We note in the following section exceptions to this. . ."

Page 4, line 10: The Jaegle et al., (2005) reference is cited for estimating uncertainty in biomass burning emissions. The GFED 3 emissions are used, so is there a more recent and appropriate estimate of uncertainty specifically for these emissions? I am not suggesting re-running the ensemble, but again (as with soil NOx - see point above) framing the choice of factor 3 uncertainty against any other estimates would be helpful.

Page 8, Line 11: OH interferences being negligible in Arctic free troposphere. Probably correct in general, but what about in biomass plumes during ARCTAS-B?

Page 8, Sec. 2.4: The detail on the specific GEOS-Chem code for aircraft flight track interpolation seems unnecessary. Instead just describe what this does.

Page 8, line 26: I am not sure you can claim that the flights give a "representative view of the Arctic troposphere". See my general point (4) above.

Page 8, line 27: You shouldn't refer to Fig. 6 before you have referred to Figs. 3,4,5. Consider re-ordering / re-numbering the figures.

Page 11, line 17: Should be "are shown in Figure 7".

Page 13, line 7-9: Mischaracterisation of advection from mid-latitudes effect on ozone. Has this been discussed in the main paper text? Previous multi-model studies have also shown low profile springtime ozone in the Arctic in GEOS-Chem, but no similar underestimation of ozone in other models driven by GEOS-5 meteorological data (e.g. Emmons et al., 2015, Figs. 16 & 17). It therefore seems unlikely to be related to advection errors. Please expand this discussion in light of this past work.

References

Arnold, S. R., Emmons, L. K., Monks, S. A., Law, K. S., Ridley, D. A., Turquety, S., Tilmes, S., Thomas, J. L., Bouarar, I., Flemming, J., Huijnen, V., Mao, J., Duncan, B. N., Steenrod, S., Yoshida, Y., Langner, J., and Long, Y.: Biomass burning influence on high-latitude tropospheric ozone and reactive nitrogen in summer 2008: a multi-model analysis based on POLMIP simulations, Atmos. Chem. Phys., 15, 6047-6068, doi:10.5194/acp-15-6047-2015, 2015.

Emmons, L. K., et al., The POLARCAT Model Intercomparison Project (POLMIP): overview and evaluation with observations, Atmos. Chem. Phys., 15, 6721-6744, doi:10.5194/acp-15-6721-2015, 2015.

Fischer, E. V., Jacob, D. J., Yantosca, R. M., Sulprizio, M. P., Millet, D. B., Mao, J., Paulot, F., Singh, H. B., Roiger, A., Ries, L., Talbot, R. W., Dzepina, K., and Pandey Deolal, S.: Atmospheric peroxyacetyl nitrate (PAN): a global budget and source attribution, Atmos. Chem. Phys., 14, 2679-2698, doi:10.5194/acp-14-2679-2014, 2014.

Mao, J., Fan, S., Jacob, D. J., and Travis, K. R.: Radical loss in the atmosphere from Cu-Fe redox coupling in aerosols, Atmos. Chem. Phys., 13, 509-519, doi:10.5194/acp-13-509-2013, 2013.

Stohl, A.: Characteristics of atmospheric transport into the Arctic troposphere, J. Geophys. Res., 111, D11306, doi:10.1029/2005JD006888, 2006.

Wespes, C., et al., Analysis of ozone and nitric acid in spring and summer Arctic pollution using aircraft, ground-based, satellite observations and MOZART-4 model: source attribution and par- titioning, Atmos. Chem. Phys., 12, 237–259, doi:10.5194/acp-12-237-2012, 2012.

---

## Referee Comment (RC2) · Anonymous Referee #2 · 15 Dec 2016

Christian et al. 2016 present a numerical sensitivity analysis employing the random sampling-high dimensional model representation (RS-HDMR) technique to understand how the uncertainty in a number of important parameters in the GEOS-Chem model affect the simulated abundances of ozone and the HOx (=OH+HO2) family in the Arctic during summer and spring. They find that the GEOS-Chem model results of ozone are fairly insensitive to the parameters that they have chosen to explore. Whilst the modelled HOx species show much greater sensitivity. Detailed analysis of the large ensemble of simulations using the RS-HDMR method identifies that the major source for model sensitivity for HO2 is owing to the large uncertainty in the uptake coefficient for HO2 onto aerosol.

[Figure]

The authors conclude that determination of the HO2 aerosol uptake coefficient remains an area for further study and that the best gamma value from their ensemble of simulations is lower than that used in the current version of GEOS-Chem.

In general this is a well written manuscript with nice clear figures. I think that the results are interesting and should stimulate some wider interest between the lab and modelling communities and recommend that this be published following appropriate response to the following concerns:

1) Number of parameters chosen: The authors state in the conclusions that 52 parameters have been explored. In Table 1 I count 51. Have I missed something? Can the authors please check this.

2) Model resolution: The current simulations have all been performed at 4ËŽ x 5ËŽ horizontal resolution with the justification that the authors found only small differences (∼10%) using higher resolution model simulations (2ËŽ x 2.5ËŽ). These latter higher resolution simulations still strike me as being very low resolution. Do the authors expect the same sensitivity to hold at say 0.5ËŽ x 0.5ËŽ? I ask as I have seen more and more simulations with GEOS-Chem at these sorts of high resolutions and so I think transferring the knowledge gained here to those studies is important.

3) Meteorological uncertainty in the model: This is out of interest, but how different is the uncertainty between the average monthly fields between GEOS-4 and GEOS-5 compared to the standard deviation of the meteorological parameters generated from the re-gridding from the native GOES-5 grid to the GEOS-Chem grid?

4) The role of organic radicals: It's interesting to see that the uncertainty in isoprene emissions pops up as having an effect on ozone and HO2. I was wondering if the authors considered the uncertainty in the organic peroxy radical reactions associated with isoprene?

5) NOx: There is very little mention of the role of NOx in the manuscript and I'm surprised that the authors did not include NOx in the analysis and results. Clearly NOx plays an important role in coupling HOx and I would like to see how the current study impacts the NOx partitioning. I think that this is something that many others would also benefit from seeing and I would suggest adding some plots to at least show the impact of the ensemble of simulations on the NOx profiles.

6) Normalised sensitivities: It's not clear to me if the reason that HO2 uptake is the most sensitive parameter is owing to the fact that it has the greatest uncertainty? Can the authors comment on the use of the method in distinguishing/determining normalised sensitivities?

Technical corrections:

Page 2 line 4: I don't think Wu et al., 2007 is a great reference for making this point. A better reference would be a multi model intercomparison study like one of the ACCMIP or HTAP papers.

Page 9 line 13: ppb should be ppt I think.

Page 9 line 22: Need to define HOx earlier in the text.

Page 10 line 10: ppb should be ppt I think.

---

## Author Comment (AC1) · 26 Jan 2017

We thank the referee for their thorough review and helpful comments. Below are our responses to the referee's comments (*italics*).

*1. It would be useful to discuss how model-dependent the large ozone sensitivity to uncertainty in the $NO_2$ + OH reaction rate may be. Previous studies comparing GEOSChem with ARCTAS observations have shown that the model displays a large overprediction of $HNO_3$ and a large under-prediction of PAN in the Arctic troposphere (Figs. 18 & 16 Emmons et al., (2015); Figs. 3 and 4 Arnold et al., 2015). To what extent is the sensitivity to the $HNO_3$ production rate a reflection of the propensity for*

*GEOS-Chem to produce large amounts of $HNO_3$? i.e. is the sink for $NO_2$ through formation of $HNO_3$ (and therefore sensitivity to uncertainty in its rate) realistic? Does this version of the model include the $NO_y$ chemistry updates from Fischer et al., (2014) which greatly improved the simulation of $NO_y$ chemistry in GEOS-Chem? The authors should include some reference to these past studies comparing GEOS-Chem with ARCTAS data and other models in the discussion, and comment on how the model Arctic $NO_y$ budget compares with observations and implications for the inferred sensitivity to the kinetic uncertainties.*

Response: In our model runs we likewise see similar over-prediction of $HNO_3$ and under-prediction of PAN in our domain. As noted, this isn't a novel result with GEOS-Chem but should be mentioned for those readers unfamiliar with the model. We've edited the manuscript to make note of this (P10 L10-15). Even with this $HNO_3$ overprediction, I'm hesitant to see it as GEOS-Chem specific result with other implementations of this method to box models in other regions finding similar sensitivity (Chen et al. 2012).

The model version used in this study (v9-02) implements many of the Fischer et al. updates such as the implementation of the Paulot isoprene oxidation scheme, updating various rate coefficients, and increasing the deposition flux of PAN. Not all of the updates suggested by Fischer et al. have been included in the standard code as of yet but are slated to be included in v11-2 http://wiki.seas.harvard.edu/geos-chem/index.php/GEOS-Chem_v11-02.

*2) The large response to soil $NO_x$ emissions is a surprising and novel result, and also warrants further discussion. Given the high vertical stability of the Arctic troposphere, there is strong isolation of the free mid / upper troposphere from emissions and processes in high latitude / Arctic boundary layer, and air tends to be transported into the mid/upper troposphere from lower latitudes (e.g. Stohl, 2006, Wespes et*

*al., 2012). Therefore ozone sensitivity at altitudes in the mid and upper troposphere is presumably driven by response to uncertainty in soil $NO_x$ emissions from lower latitudes, and its impacts on ozone chemistry during uplift and long-range transport into the Arctic? It would be useful to expand on this in the manuscript, such that the reader has a better idea of what drives this sensitivity. A factor 3 uncertainty is assumed for these emissions based on Jaegle et al., (2005). Is this the most appropriate and recent reference for framing this uncertainty? Given the importance of this uncertainty for ozone in the N American Arctic, it would be helpful to discuss more widely estimates of the reliability (uncertainty in) the soil emissions if other studies are available and how robust the factor 3 estimate may be.*

Response: Over much of our Arctic domain in the summer, soils along with biomass burning are the primary emissions sources of $NO_x$ in the model because of the lack of major anthropogenic sources. The stability of the Arctic atmosphere brought up in the Stohl and Wespes et al. papers is more of an issue for the winter and spring periods in which the thermal inversion is stronger. In the case of the Stohl paper, the greatest summertime sensitivity to midlatitude transport was further north than almost all the flights in ARCTAS-B. Also, in our results the sensitivity to soil $NO_x$ was most pronounced in the summertime, not the spring when this higher altitude transportation from the mid-latitudes is more important over the Arctic domain. You are correct that advection from the midlatitudes into the mid-high troposphere is an important consideration in this domain, especially for the springtime. This point was made noting the sensitivity to Asian and USA emissions P10 L20-24. Bringing up specifically the dynamic reasons for this sensitivity is a good idea and is now made more explicitly (P10 L 24).

As far as the chosen uncertainty range, you are correct in there being some uncertainty to our chosen uncertainties. In the case of soil $NO_x$, there has been some more recent efforts made with satellite data such as Vinken et al. (2014) to reduce this uncertainty. However, as they note in citing Schumann and Huntrieser (2007), there is still a large variability in these estimates (4-15 TgN yr$^{-1}$). This large range of estimates carries over to biomass burning emissions as well (6-12 TgN yr$^{-1}$) (also Schumann and Huntrieser, 2007 as cited by Vinken et al., 2014). With this, a factor of 3 uncertainty may be slightly on the high side, but not unreasonable in our opinion. In tests we also varied the uncertainty of all the factors to $\sigma/2$ and $2\sigma$ in addition to the $1\sigma$ analyzed in this study and found almost exactly the same qualitative results (quantitatively the sensitivity indices values varied a few percent) giving us confidence in these results for a variety of different uncertainty ranges.

Changes: Added reference to the Vinken et al. and Schumann and Huntrieser papers P4 L10-13.

*3) To what extent is the large HO$_x$ response to gamma HO$_2$ a reflection of the large uncertainty range implemented (factor 3)? It would be useful to show what actual range of gamma values this corresponds to. The authors show that the ensemble members with lower gamma values best match profile observations of HO$_2$. How do these gamma HO$_2$ values compare with those used in previous GEOS-Chem studies? What are the implications for model comparisons with high latitude CO values, which in previous studies have been improved by implementing different formulations of aerosol uptake of HO$_2$ (e.g. Mao et al., 2013)? How does the choice of product (H$_2$O$_2$ or H$_2$O) affect comparisons with CO and ozone? It would be useful to discuss this, since underestimation of CO at high latitudes in CTMs is a persistent problem (e.g. Emmons et al., 2015).*

Response: Certainly the high uncertainty in Gamma HO$_2$ contributes to the high sensitivity. This high uncertainty is both evident in the JPL evaluation and in the wide range of treatment and values historically used in GEOS-Chem (P5 L10-17). Also, as we noted in the response to the previous point (# 2), in tests varying the uncertainty

ranges, we found very similar results.

We described on Page 7 how we constructed the distributions in Section 2.2.1 ("Uncertainties"). As the perturbations followed a lognormal distribution, listing a range of values may not be most useful to the readers as the high and low values would be in the tails of the distribution and not indicative of the vast majority values used in the study. Excluding the upper and lower 5% of the distribution, the values roughly range from 0.04 to 1 which is within the range of values historically used in GEOS-Chem. We touched on the range of gamma values in (P12 L26, P13 L26) describing what values of gamma $HO_2$ provided the closest match to observed summertime $HO_2$ profiles.

As for CO, when the modeled $HO_2$ uptake produces $H_2O_2$ instead of $H_2O$, we find CO mixing ratios to be decreased throughout the vertical profile on the order of 10ppb for both spring and summer. Thus, this change exasperates the underprediction of modeled CO with the uptake product of gamma $HO_2$ being $H_2O_2$ rather than $H_2O$. As you note, models tend to underestimate CO in the high latitudes. While this is the case for the Arctic spring, in the summer we found the model to over-estimate CO by around a factor of 2 in the lowest 2km of the troposphere before shifting to under-prediction above 4km (Figs S1 & S2). As for ozone, we found very modest differences between these two scenarios as evidenced by the blue dashed lines in (Figs 2 & 6).

Changes: For readers interested in CO profiles and how the aerosol uptake product of $HO_2$ affects CO profiles we've created figures for both spring and summer in a new supplement (S1, S2).

*4) It should be made clear in the abstract and the methodology that this analysis only provides information on drivers of model response to uncertainties in air masses sampled during ARCTAS. It cannot be assumed that this is representative of the whole Arctic unless this can be shown explicitly. Figure 2 shows a good spread of aircraft observations across altitudes, but the flights still only sample the N American Arctic on*

*specific days, when there are certain specific air mass origins.*

Response: This is a good point. While when writing the paper we thought readers would understand the geographic limitation of the study area, but it is probably best to make it clearer as suggested.

Changes: In the abstract instead of "period", "flight tracks" is substituted (P1 L6). Also P8 L30-31 changed to "...providing a fairly representative view of the Arctic troposphere over this domain for the times corresponding to these flights."

Specific / minor comments

*Page 1, Line 1: "oxidation capability" change to "oxidation capacity"*

Changed as suggested

*Page 1, Line 19/20: "Increasing oil and gas exploration and extraction, coupled with summertime shipping lanes through the region will make air pollution worse". This statement needs a reference.*

Changes: Added a citation to Granier et al. 2006.

*Page 2, line 5: ".. model shortcomings are usually attributed to errors in the chemical reaction rates, emissions, or meteorology (e.g., Wild and Prather, 2006)". The cited study is specifically about effects of model resolution? Please cite examples to back up the specific reasons you list.*

Changes: The Wild and Prather paper made this point (section 3, paragraph 13). In lieu of this general point, we've added citations to some papers dealing with each of these three specifically (meteorology, emissions, chemistry) Kinnison et al. 2007 for meteorology, Fischer et al 2014/Jaegle et al. for emissions, Chen et al., 1997 for chemical reaction rates.

*Page 2, Line 10: Omit semi-colon.*

Changed as suggested

*Page 2, line 13: "two more input factors" should be "two or more input factors"?*

Correct. Changed as suggested.

*Page 4, line 3: Better phrased as: "We note in the following section exceptions to this. . ."*

Changed as suggested

*Page 4, line 10: The Jaegle et al., (2005) reference is cited for estimating uncertainty in biomass burning emissions. The GFED 3 emissions are used, so is there a more recent and appropriate estimate of uncertainty specifically for these emissions? I am not suggesting re-running the ensemble, but again (as with soil $NO_x$ - see point above) framing the choice of factor 3 uncertainty against any other estimates would be helpful.*

See comments for general point 2

[Figure]

*Page 8, Line 11: OH interferences being negligible in Arctic free troposphere. Probably correct in general, but what about in biomass plumes during ARCTAS-B?*

When excluding OH measurements taken within smoke plumes (HCN > 1000 ppt), the mixing ratios differ less than 10% in nearly all the vertical bins. This is similar to what was noted in the paper with $HO_2$ where there is a similarly small effect. This difference doesn't change the conclusions of the paper.

*Page 8, Sec. 2.4: The detail on the specific GEOS-Chem code for aircraft flight track interpolation seems unnecessary. Instead just describe what this does.*

Scaled back a bit P8 L18-25 and removed the last sentence in that section.

*Page 8, line 26: I am not sure you can claim that the flights give a "representative view of the Arctic troposphere". See my general point (4) above.*

Response to general point 4 should cover this.

*Page 8, line 27: You shouldn't refer to Fig. 6 before you have referred to Figs. 3,4,5. Consider re-ordering / re-numbering the figures.*

The order of the figures seems to be in a good, logical order as currently ordered so the reference to these figures has been removed here. The new sentence was already edited for general point 4.

*Page 11, line 17: Should be "are shown in Figure 7".*

Correct. Changed as suggested

*Page 13, line 7-9: Mischaracterisation of advection from mid-latitudes effect on ozone. Has this been discussed in the main paper text? Previous multi-model studies have also shown low profile springtime ozone in the Arctic in GEOS-Chem, but no similar underestimation of ozone in other models driven by GEOS-5 meteorological data (e.g. Emmons et al., 2015, Figs. 16 & 17). It therefore seems unlikely to be related to advection errors. Please expand this discussion in light of this past work.*

Response: Thank you for bringing this recent literature to our attention. After considering some of the $NO_x$ profiles, the ozone conclusions have been refocused in a different direction and mention the POLMIP results.

Changes: Moved discussion of Alvarado paper and its comparison to the POLMIP results into Section 3.2.2 (P11, L10-14)

---

## Author Comment (AC2) · 26 Jan 2017

gensymb We thank the referee for their thorough review and helpful comments. Below are our responses to the referee's comments (*italics*).

*1. Number of parameters chosen: The authors state in the conclusions that 52 parameters have been explored. In Table 1 I count 51. Have I missed something? Can the authors please check this.*

There were 52 factors included in the HDMR analysis and 51 in the table. EPA $NH_3$

should have been included in the table and has now been added. Thank you for finding this discrepancy.

*2) Model resolution: The current simulations have all been performed at 4° x 5° horizontal resolution with the justification that the authors found only small differences (∼10%) using higher resolution model simulations (2° x 2.5°). These latter higher resolution simulations still strike me as being very low resolution. Do the authors expect the same sensitivity to hold at say 0.5° x 0.5°? I ask as I have seen more and more simulations with GEOS-Chem at these sorts of high resolutions and so I think transferring the knowledge gained here to those studies is important.*

Response: We would have preferred to run this analysis at the finest possible resolution but are limited by computational resources. The comparison to the 2° x 2.5° was intended more to illustrate how sensitive the modeled results are to changes in resolution. This comparison between 4° x 5° and 2° x 2.5° has been used in previous GEOS-Chem studies (eg., Fiore et al., 2002, Fischer et al., 2014). The expectation is that most of the findings of this paper would be applicable to other model resolution choices considering the small differences between the two resolutions we tested. While finer resolution (like 0.5° x 0.5°) studies are becoming more popular with GEOS-Chem, these studies are limited to a few regions–E Asia, Europe, and North America. In the case of the North American domain, the nested grid doesn't cover the ARCTAS domain.

*3) Meteorological uncertainty in the model: This is out of interest, but how different is the uncertainty between the average monthly fields between GEOS-4 and GEOS-5 compared to the standard deviation of the meteorological parameters generated from the re-gridding from the native GOES-5 grid to the GEOS-Chem grid?*

Response: By averaging over the month, some of the day-to-day and some of the spatial differences are muted between the resolution choices. From some back of the envelope calculations between the 2° x 2.5° and 4° x 5° resolution meteorological fields, we find differences less than those between meteorological models with the greatest differences coming around the edges of mountainous regions and the edge of the Antarctic continent. Going even finer to the native resolution would presumably further increase these differences to being around the same or perhaps greater than the differences between the GEOS-4 & 5 models.

*4) The role of organic radicals: It's interesting to see that the uncertainty in isoprene emissions pops up as having an effect on ozone and $HO_2$. I was wondering if the authors considered the uncertainty in the organic peroxy radical reactions associated with isoprene?*

Response: All the chemical reactions in the GEOS-Chem chemical mechanism were included in the Morris Method pre-screen test including those involving organic peroxy radicals. The isoprene peroxy radical reactions did not make the cut to be included in the HDMR analysis but some of the methane ones did as shown in Table 1.

*5) $NO_x$: There is very little mention of the role of $NO_x$ in the manuscript and I'm surprised that the authors did not include $NO_x$ in the analysis and results. Clearly $NO_x$ plays an important role in coupling $HO_x$ and I would like to see how the current study impacts the $NO_x$ partitioning. I think that this is something that many others would also benefit from seeing and I would suggest adding some plots to at least show the impact of the ensemble of simulations on the $NO_x$ profiles.*

Response: $NO_x$ profiles were not originally included in the paper for a couple of reasons. First, this analysis didn't change the model treatment of $NO_x$ (except for the perturbations to emissions and chemical rates) making most of that analysis a rehash

of previous research. Secondly, a few different $NO_x$ emissions inventories were perturbed in the analysis leading to a large variability in their modeled concentrations between model runs, especially near the surface where the emissions sources are. As you note though, readers would be interested in the seeing at least the $NO_x$ profiles. To address this, we have created a small supplement showing the median $NO_x$ and CO profiles for both the spring and summer flights.

Changes: We've made plots showing NO, $NO_2$, and CO profiles for both ARCTAS A and B in a supplementary file (Figures S1 and S2).

*6) Normalised sensitivities: It's not clear to me if the reason that $HO_2$ uptake is the most sensitive parameter is owing to the fact that it has the greatest uncertainty? Can the authors comment on the use of the method in distinguishing/determining normalised sensitivities?*

Response: As far as the $HO_2$ uptake uncertainty, please refer to my response to referee # 1, general point # 3.

The HDMR method is not necessarily used to determine normalized sensitivities, however one could infer a qualitative sense of this comparing those factors in the pie charts to their respective sensitivities listed in Table 1. We touched on this in a peripheral sense noting the sensitivity of the oxidants to both the chemical kinetic rates (which have much lower uncertainties) and emission inventories (P3, L20-24). Due to the non-linearity of the chemical system, I have reservations with creating "normalized" sensitivity indices armed with the sensitivity indices and uncertainty factors though.

Technical corrections:

*Page 2 line 4: I don't think Wu et al., 2007 is a great reference for making this point. A*

*better reference would be a multi model intercomparison study like one of the ACCMIP
or HTAP papers.*

Response: That's a good suggestion to use a multi model inter-comparison paper to
make this general point. Instead of the ACCMIP or HTAP papers, we've edited this
reference to the POLMIP paper as it's also Arctic focused.

Changed the reference to Emmons et al., 2015

*Page 9 line 13: ppb should be ppt I think.*

Correct. Changed as suggested

*Page 9 line 22: Need to define HO$_x$ earlier in the text.*

HO$_x$ was defined on Page 2, Line1. No changes

*Page 10 line 10: ppb should be ppt I think.*

Correct. Changed as suggested